# *Phaseolus vulgaris Erythroagglutinin* (PHA-E)-Positive Ceruloplasmin Acts as a Potential Biomarker in Pancreatic Cancer Diagnosis

**DOI:** 10.3390/cells11152453

**Published:** 2022-08-08

**Authors:** Shanshan Sha, Yating Wang, Menglu Liu, Gang Liu, Ning Fan, Zhi Li, Weijie Dong

**Affiliations:** 1College of Basic Medical Sciences, Dalian Medical University, Dalian 116044, China; 2Clinical Laboratory, Dalian Municipal Central Hospital, Dalian 116033, China

**Keywords:** pancreatic cancer, glycoproteins, bisecting GlcNAc, biomarker, ceruloplasmin

## Abstract

Pancreatic cancer (PC) remains one of the top 10 causes of cancer-related death in recent years. Approximately 80% of PC patients are diagnosed at the middle or advanced stage and miss the opportunity for surgery. The demand for early diagnostic methods and reliable biomarkers is increasing, although a number of tumor markers such as CA19-9 and CEA have already been utilized in clinics. In this study, we analyzed the alteration of *N*-glycan of serum glycoproteins by mass spectrometry and lectin blotting. The results showed that bisecting GlcNAc structures of glycoproteins are significantly increased in PC patients’ sera. With *Phaseolus vulgaris Erythroagglutinin* (PHA-E) lectin that specifically recognizes bisecting GlcNAc *N*-glycans, the serum glycoproteins bearing bisecting GlcNAc in PC patients’ sera were pulled down and identified by nano-LC-MS/MS. Among them, ceruloplasmin (Cp) was screened out with a satisfied sensitivity and specificity in identifying PC from acute pancreatitis patients (AUC: 0.757) and normal healthy persons (AUC: 0.972), suggesting a close association between Cp and PC development and diagnosis. To prove that, the Cp expression in tumor tissues of PC patients was examined. The results showed that Cp was significantly upregulated in PC tissues compared to that in adjacent normal tissues. All these results suggested that PHA-E-positive Cp could be a potential PC-specific glycoprotein marker to distinguish PC patients from acute pancreatitis patients and normal persons.

## 1. Introduction

Pancreatic cancer (PC) is the seventh leading cause of cancer death in both sexes, with a mortality of 4.7% worldwide in 2020 [1]. Its poor survival rate has not been improved in nearly 40 years. In some countries, the incidence and mortality of PC have been slightly increased probably because of the increasing prevalence of obesity, diabetes, and alcohol consumption. It is predicted that PC will surpass breast cancer and become the third leading cause of cancer death by 2025 in European countries [1].

The difficulty of early diagnosis is one of the main reasons for the high mortality of PC. Because no obvious symptoms are presented at the early stage of this disease, most patients are diagnosed at the middle or advanced stage of PC and miss the best time to achieve an optimal treatment [2]. Surgical resection is the only curative treatment for PC [3]; however, only 9.7% of PC patients have no tumor metastasis at the time of diagnosis and only 15% to 20% of PC patients can be candidates for surgery due to poor early diagnosis [4]. A lack of reliable and specific biomarkers remains the major challenge for PC. Although a number of tumor markers, including carbohydrate antigen 19-9 (CA19-9), CA72-4, CA50, CA242, carcinoembryonic antigen (CEA), macrophage inhibitory cytokine 1 (MIC 1), and CEA-related cell-adhesion molecule 1 (CEACAM1), have been evaluated for PC diagnosis, none of them are proven to have sufficient diagnostic accuracy. Among them, CA19-9 is the only marker approved by FDA and widely employed clinically [5]. However, it is not reliable enough for its low sensitivity and specificity in the early detection of PC [6]. Besides PC, CA19-9 is also expressed highly in other malignant tumors and diseases, such as gastric cancer, colorectal cancer, and liver cirrhosis [7,8]. The positive predictive value of CA19-9 in PC diagnosis is only 0.5–0.9% [4]. More importantly, 5–10% of the population are Lewis-antigen-negative and unable to produce CA 19-9 [9]. On the other hand, imaging techniques such as ultrasonography, computed tomography, and magnetic resonance imaging are also commonly helpful for the diagnosis of PC. Yet, these diagnostic methods are difficult to make early diagnosis due to the native morphology of PC, such as hidden anatomical location, small size, and desmoplastic characteristics [10,11]. Therefore, developing efficient biomarkers or tools for early diagnosis is urgently required for PC patients.

Disease-specific glycoproteins, which have glycoalterations in particular glycoproteins and can enhance the diagnostic sensitivity and specificity, attract much attention in biomarker developments in recent years [12]. Glycosylation is one of the most important post-translational modification of proteins, and abnormal glycosylation is a universal feature of most human cancers. The differential expressed sugar chains or special glycosylation structures have been a hallmark of malignancy associated with tumor proliferation, angiogenesis, migration, and invasion [13,14,15]. It was reported that bisecting *N*-GlcNAc modification on EGFR and small extracellular vesicles can diminish malignancy of breast cancer cells, which may contribute to the development of novel targets in breast cancer therapy [16,17]. In terms of disease diagnosis, quite a few of biomarkers, such as AFP- L3 for hepatocellular carcinoma (HCC), MUC1 (CA15-3) for breast cancer, MUC16 (CA125) for ovarian cancer, WFA-MUC1 for cholangiocarcinoma, and PSA for prostate cancer, are also developed in dependence of the alterations of glycans on the core protein [18,19,20,21]. The difference in glycan expression combined with the change in protein level makes these molecules optimal biomarkers for the diagnosis of specific diseases. Thus, it will bring hope for PC diagnosis if PC-specific glycoproteins can be identified.

Ceruloplasmin (Cp) is known as a glycoprotein associated with malignancy in many cancers, including lung cancer and breast cancer [22]. A previous study using electrospray ionization (ESI) iontrap tandem mass spectrometry (MS) revealed that serum Cp was elevated in the sera of PC patients as compared with normal volunteers [23], suggesting that Cp could be a potential biomarker for PC. In terms of the glycosylation, it was reported that an increased sialyl-Lewis X on Cp was presented in the sera of PC patients [24]. However, so far, the relationships between the bisecting GlcNAc structures of sera glycoproteins and pancreatic cancer, and the role of bisecting GlcNAc structures of Cp in the development of PC, have not been reported.

In this study, the expression of various types of *N*-glycan in PC patients’ sera were analyzed and compared with those in the sera of acute pancreatitis (AP) patients and normal healthy individuals (normal controls, NC) by lectin blotting and MS. The results showed that bisecting GlcNAc structures of glycoproteins are significantly enhanced in the sera of PC patients. Serum glycoproteins with bisecting GlcNAc *N*-glycans were pulled down by PHA-E lectin in PC patients’ sera and identified by nano-LC-MS/MS. Among identified glycoproteins, we found that the expression of Cp and apolipoprotein E (Apo-E) in PC patients was significantly higher than those in AP patients and NC individuals after PHA-E pull-down. These data provide a theoretical basis for the diagnosis of PC patients and further exploration of the deep mechanism of Cp in PC progression.

## 2. Materials and Methods

### 2.1. Samples

The serum of AP and PC patients and NC individuals were obtained from Dalian Municipal Central Hospital affiliated to the Dalian Medical University, China. The clinical characteristics of these patients were presented in Appendix A. All patients were diagnosed by pathological examinations and the specimens were obtained prior to surgery or biopsy and stored at −80 °C until use. Informed consents were signed by all patients and all research protocols in this study were conducted with the approval of the Ethical Committee and Institutional Review Board of Dalian Municipal Central Hospital (YN2021-005-01). For some experiments, equal quantities of sera were extracted from 8 samples of one group (labeled with asterisks in Appendix A) and mixed together to obtain pooled samples.

### 2.2. Lectins and Antibodies

Biotinylated lectins, including *Aleuria aurantia* lectin (AAL), *Lens culinaris* agglutinin (LCA), PHA-E, *Phaseolus vulgaris Leucoagglutinin* (PHA-L), and *Sambucus nigra* agglutinin (SNA) (Appendix A), were purchased from Vector Laboratories Inc. (Burlingame, CA, USA). Rabbit anti-Cp, anti-transferrin (Tf) and anti-Apo-E polyclonal antibodies were purchased from Boster (Wuhan, China). Horseradish peroxidase (HRP)-labeled anti-rabbit IgG antibody and streptavidin-HRP were obtained from Beyotime (Shanghai, China). Streptavidin-immobilized agarose beads (Invitrogen, Carlsbad, CA, USA) were purchased from Thermo Fisher Scientific (Waltham, MA, USA).

### 2.3. Purification of N-Glycans from Serum for Mass Spectrometry (MS) Analysis

Pooled serum samples (*n* = 8, 300 μg) were dissolved in 30 μL of 80 mM ammonium bicarbonate containing 10 mM DL-dithiothreitol at 60 °C for 30 min, followed by adding 2.2 μL of 250 mM iodoacetamide and incubating at room temperature (RT) for 60 min in the dark. Then, the serum proteins were treated with 0.05 mg/mL trypsin (Sigma-Aldrich, St. Louis, MO, USA) at 37 °C for 18 h. After heat-inactivation by trypsin at 95 °C for 5 min, the peptides were treated with 3 mU peptide *N*-glycanase F (PNGase F) (Takara Bio Inc., Dalian, China) at 37 °C for 18 h and the generated whole-serum *N*-glycans were purified using the BlotGlyco^®^ glycan purification kit according to the manufacturer’s protocol (Sumitomo Bakelite Co., Tokyo, Japan). Briefly, the glycans were captured by BlotGlyco^®^ beads and methylesterified of sialic acids with 3-methyl-1-*p*-tolytriazene (98%, Sigma-Aldrich). Then, the glycans were labeled and released with an aminooxy-functionalized peptide reagent (aoWR), and the derivatized glycans were recovered from the resin by adding 50 μL distilled water. Finally, the excess reagents in the derivatized glycans were removed by using a cleanup column provided in the kit. The obtained solution containing the glycan derivatives was used for the following MS analysis.

### 2.4. Lectin and Western Blotting

The protein concentration of serum was determined by the bicinchoninic acid (BCA) protein assay kit (Takara Bio Inc.). For electrophoresis, serum proteins were incubated with SDS-PAGE sample buffer (125 mM Tris-HCl, pH 6.8, 4% SDS, 20% glycerol, 10% beta-mercaptoethanol, and 0.004% bromophenol blue) at 100 °C for 5 min, loaded in the gel and separated in Tris-glycine SDS running buffer (25 mM Tris, 250 mM glycine, 0.1% SDS). After electrophoresis, the proteins were transferred onto the Immobilon-P PVDF membrane (Millipore, Billerica, USA) at 80 mA for 1 h by using Tris-glycine SDS (48 mM Tris, 39 mM glycine, 0.037% SDS) transfer buffer with 20% methanol. To maximize the number of proteins retained on electroblotted PVDF membranes, a fixation method was performed as we reported recently [25]. Briefly, for lectin blotting, the electroblotted PVDF membrane was immersed in acetone at RT for 30 min, followed by heating at 100 °C for 30 min. For Western blotting, the electroblotted PVDF membrane was immersed in 0 °C acetone for 30 min, followed by heating at 50 °C for 30 min.

For lectin blotting, the fixed PVDF membranes were blocked with 5% BSA for 60 min and then washed with TBST solution (20 mM Tris, 150 mM NaCl and 0.05% Tween 20) 3 times (5 min each time), and finally incubated with biotinylated lectins at 4 °C overnight. The lectins were diluted in TBST at the work concentration of 0.13 μg/mL for AAL, 0.33 μg/mL for LCA, 0.1 μg/mL for PHA-E, 2 μg/mL for PHA-L, and 0.13 μg/mL for SNA. After being washed with TBST 3 times, the membranes were incubated with TBST diluted streptavidin-HRP for at RT 1 h. The bands were visualized using the ECL (enhanced chemiluminescence) Plus reagents (Beyotime). The protocol for immunoblotting was similar with that for lectin blotting, except for the antibody concentrations, membrane washing conditions, and the incubation time of secondary antibody. All primary antibodies were diluted in TBST at the ratio of 1:1000. After the incubation of primary antibodies, the membranes were washed with TBST for four times (10 min each time), incubated with the secondary antibody at RT for 2 h, and finally visualized by ECL Plus reagents.

For each lectin blotting and immunoblotting, chemiluminescent signals were recorded using a ChemiDoc MP imaging system (Bio-Rad, Hercules, CA, USA), and Image J software was used to quantify the protein bands. The total protein signal in Coomassie brilliant blue (CBB) staining of each sample was chosen as a standard control. The relative intensity was calculated by dividing the signal of each line in lectin blotting or a band in Western blotting with the total protein signal of corresponding line in CBB staining.

### 2.5. Lectin Pull-Down Assay

To improve the pull-down efficiency of the proteins that contain bisecting GlcNAc by PHA-E lectin, we first removed the albumin, which is the most abundant serum protein according to previous reports [26]. Total serum protein (20 μL) was mixed with 30 volumes of cold trichloroacetic acid (TCA)/acetone (10% TCA in analytical grade acetone) and incubated at −80 °C for 2 h. The protein precipitate was then collected by centrifugation at 10,000× *g* for 20 min at 4 °C. After being washed with 1 mL acetone in ice for 15 min and centrifuged again, the proteins were resolved in 2 M urea and their concentrations were determined using the BCA protein assay kit. For lectin pull-down, 350 μg proteins were incubated with biotin-PHA-E lectin at 4 °C overnight. Then, the streptavidin-agarose was added to the mixture and incubated at 4 °C for 3 h, and the captured proteins were analyzed with 10% SDS-PAGE. The bands of proteins that were differently expressed among NC, AP and PC groups were finally extracted from the SDS gel and identified by LC-MS/MS.

### 2.6. In-Gel Protein Digestion

The gel bands were cut into smaller pieces and transferred into microtubes. After washed with Milli-Q water (Millipore), the gels were destained with 50 mM ammonium bicarbonate dissolved in acetonitrile (Sigma-Aldrich). After completely destaining and residual detergents removing, the samples were dehydrated in acetonitrile. When the gel pieces became opaque white, they were reduced and alkylated, respectively. Next, the gel pieces were digested for 16 h by trypsin (Sigma-Aldrich) at 37 °C. The peptide mixtures were collected and transferred into new microtubes for further analysis by Michrom AdvanceTM nano/cap LC-Q-TOF MS (Bruker, Auburn, CA, USA).

### 2.7. Identification of Proteins via Nano-LC-MS/MS

From the peptide mixtures, 5 μL solution was initially applied to a 2 cm long (100 μm internal diameter) trap column packed with 5 μm, 200 A Magic C18 AQ matrix (Michrom Bioresources, Auburn, CA, USA) followed by separation on a 15 cm long column that was packed with 3 μm, 200 A Magic C18 AQ. Samples were loaded onto the trap column at 10,000 nL/min, and chromatographic separation occurred at 200 nL/min. Mobile phase A consisted of 0.1% (*v*/*v*) formic acid in water, while mobile phase B consisted of 0.1% (*v*/*v*) formic acid in acetonitrile, and gradient conditions were as follows: 5 to 40% B in 80 min, up to 80% B in 4 min, maintaining this concentration for 10 min. Eluted peptides were directly introduced into a CaptiveSpray Ionization (CSI)-Q-TOF MS (Bruker, Billerica, MA, USA) for analysis. Dry temperature to 165 °C and capillary voltage to 1500 V. MS1 spectra were acquired from 50 to 2200 *m*/*z* at about 20,000 resolution (for *m*/*z* 445.1200). For each spectrum, the five most intense ions were subjected to collision-induced dissociation (CID) fragmentation, followed by MS2 analyzer.

All MS/MS spectra were searched using Compass 1.4, Data Analysis 4.1 build 335 (Bruker, Billerica, MA, USA), and Proteinscape 3.0 (Bruker, Billerica, MA, USA) to identify peptides. Peak lists of raw files were generated by using Mascot Daemon (Matrix Science, London, UK) and searched against the Swiss-Prot or NCBI protein databases by using the search program Mascot server.

### 2.8. Immunohistochemistry (IHC)

Formalin-fixed and paraffin-embedded tissue section samples (PC tissue and adjacent normal pancreatic tissue) were prepared. Then, these sections were deparaffinized with xylene, rehydrated by alcohol gradient, followed by antigen retrieval by sodium citrate. The sections were incubated in 3% hydrogen peroxide (H_2_O_2_) solution and blocked with 5% fetal goat serum. Next, the sections were incubated with rabbit polyclonal antibody, anti-Cp (1:300 dilutions, Proteintech, Wuhan, China) at 4 °C overnight. After washing with PBS, sections were incubated with a biotinylated secondary antibody at 37 °C for 1 h. Subsequently, sections were treated with diaminobenzidine (DAB; Beyotime, Nantong, China), stained with hematoxylin and dehydrated using gradient alcohol. Finally, an Aperio Scanscope XT (Leica Biosystems, Vista, CA, USA) was used to digitally scan the slides. IHC score counting was performed by ImmunoRatio (Image J software 1.5.2, National Institutes of Health, Bethesda, USA) to prevent manual or interobserver bias. It was applied to calculate the percentage of positively stained nuclear area (labeling index) by using a color deconvolution algorithm for separating the staining components (DAB and hematoxylin) and adaptive thresholding for nuclear area segmentation. Every digital image was quantitatively analyzed in three independent areas.

### 2.9. Expression of Cp and N-Acetylglucosaminyltransferase III (GnT III) in PC from GEO Datasets

The expression of Cp and GnT III in PC and adjacent normal tissues was obtained from databases of GSE28735 database (http://www.ncbi.nlm.nih.gov/geo, accessed on 15 March 2022), GEPIA (http://gepia.cancer-pku.cn/detail.php, accessed on 2 July 2022) and BBCancer (http://bbcancer.renlab.org/, accessed on 2 July 2022). Their difference between PC and adjacent normal tissues was compared by paired *t*-test.

### 2.10. Statistical Analysis

The data were expressed as mean ± standard deviation (SD) from 8 or 12 individuals of one group, or from the triple tests of each group. Graphpad Prism 6.01 software (GraphPad Software, Inc., San Diego, CA, USA) was used for statistical analysis and one-way ANOVA was selected to determine the significance of differences among the examined groups. *p*< 0.05 was considered to be statistically significant.

## 3. Results

### 3.1. Altered Glycan Patterns of Glycoproteins in PC Patients’ Sera

To investigate the difference of *N*-glycan patterns among NC persons and AP and PC patients, the serum glycoproteins from 24 individuals, comprising 8 NC persons, 8 AP patients, and 8 PC patients (Appendix A) were detected with various lectins (Appendix A). The results showed that five types of *N*-glycans, fucose, and α1-6-linked fucose residues that are recognized by AAL and LCA lectin, respectively; bisecting GlcNAc and multiantennary *N*-glycans that are bound by PHA-E and PHA-L, respectively; and α2-6-linked sialic acid residues that are recognized by SNA, were significantly increased in PC sera as compared with those in NC and AP sera (Figure 1a, the exact values of relative intensity can be seen in Appendix A). It suggested that these serum *N*-glycans or serum glycoproteins containing these *N*-glycans can be used as biomarkers to screen PC patients.

To facilitate the subsequent experiments of glycomics, we performed lectin blotting analyses on the pooled samples of each group by above five lectins and found that the alterations in glycosylation from pooled samples were in accordance with those detected in individual samples (Figure 1b). In this way, we used the pooled samples of each group for the following the MS analysis of *N*-glycan types, PHA-E pull-down experiment and the MS analysis of pulled-down glycoproteins.

### 3.2. Altered N-Glycan Types in PC patients’ Sera

To confirm the glycosylation alterations in the serum proteins of PC patients, *N*-glycans were extracted from the pooled serum proteins of NC, AP, and PC groups and analyzed by MS (Figure 2a). All glycan signals in MS spectra were summarized in Table 1 and relative intensities of all *N*-glycan types in different groups that compared with the internal standard glycan were shown as sunburst charts (Figure 2b), along with the estimated distribution of different *N*-glycan types (Figure 2c). In total, 40 kinds of glycoforms (No. 1–40) were identified in the control and patients’ sera. Compared with the NC group, the relative abundances of bisecting glycans, mannosylation, and fucosylation were significantly increased in the PC group; compared with the AP group, the relative abundances of bisecting glycans and fucosylation were significantly enhanced in the PC group; however, the relative abundances of fucosylation seemed similar between the NC and AP groups. Together, the results of lectin blotting and mass spectrometry analysis revealed that the bisecting glycans were dramatically upregulated in PC patients and could distinguish PC patients from NC group (*p* < 0.001) and AP patients (*p* < 0.05).

### 3.3. Pull-Down Glycoproteins with PHA-E Lectin

Since glycans are difficult to be developed as biomarkers alone, we analyzed the serum glycoproteins containing bisecting glycans among three groups by pulling down with biotinylated PHA-E lectin and streptavidin agarose beads. The extracted PHA-E binding proteins were analyzed by SDS-PAGE (Figure 3a). The differently expressed protein bands in PC patients (Figure 3a, indicated by the red frame) were excised and digested in gel, and subsequently analyzed by nano-LC-MS/MS. The obtained data of identification of each band were shown in Table 2. Among these proteins, we noticed that three proteins, Cp, Apo-E, and Tf, had relative higher protein scores according to nanoLC-MS/MS analysis and their molecular weights (MWs) were well-corresponded with the bands in the gel (Figure 3a). Cp, Apo-E, and Tf were reported as pivotal oncogenic proteins involved in the progression of PC and have the potential for PC diagnosis [22,23,27,28]. Therefore, we further analyzed the level of Cp, Apo-E, and Tf in PHA-E pulled-down serum proteins by using their specific antibodies though Western blotting. The results (Figure 3b, Appendix A) showed that the expression of Cp, Apo-E, and Tf were all increased in the AP and PC groups compared to the NC group, and Cp and Apo-E were significantly higher in the PC group than in AP group (*p* < 0.01).

### 3.4. The Levels of PHA-E-Positive Cp and Apo-E in PC Patients’ Sera

Now, we have identified two candidates for PC diagnosis: serum Cp and Apo-E. However, it is unclear which is more suitable as a biomarker between Cp and Apo-E and PHA-E-positive Cp and Apo-E. To this end, the levels of Cp and Apo-E both in serum proteins and PHA-E pulled-down serum proteins from 12 NC individuals, 12 AP patients and 12 PC patients (Appendix A) were detected by Western blotting. The same amount of serum proteins from NC, AP, and PC groups was loaded onto SDS-PAGE (Figure 4a and Appendix A) and then Western blotting analysis was performed using anti-Cp and anti-Apo-E antibodies. The results showed that as an acute-phase reactive protein, Cp was significantly upregulated in AP and PC patients as compared to the NC group (Figure 4b, Appendix A). Similar results were obtained in the analysis of Apo-E (Figure 4b, Appendix A). In ROC curves, the area under ROC curve (AUC) values were more than 0.9 between the NC and AP groups and between the NC and PC groups, but only 0.576 (Cp) and 0.535 (Apo-E) between the PC and AP groups (Figure 4c), which indicated an unsatisfactory predictive capacity of Cp and Apo-E in differential diagnosis of PC from AP patients. However, the relative concentration of Cp in the PC group was remarkably increased after PHA-E pull-down. Significant differences were observed not only between the PC and NC groups (*p* < 0.01), but also between AP and PC samples (*p* < 0.05) (Figure 4d, Appendix A). The AUC for Cp between NC and AP, NC and PC, and PC and AP, were 0.917, 0.972, and 0.757, respectively. The AUC value for Apo-E was also improved after PHA-E pull-down treatment (0.611), but no significant difference was found between the PC and AP groups either (Figure 4e). All these results suggested that PHA-E-positive Cp could be taken as a potential biomarker for PC diagnosis.

### 3.5. The Expression of Cp and GnT III in PC Tumor Tissues

To confirm the role of Cp in PC tumorigenesis, we detected the level of Cp in tumor tissues and adjacent noncancerous tissues from 3 PC patients by IHC staining. The results showed that the expression of Cp in tumor tissues was significantly higher than that in paired adjacent noncancerous tissues from the same patient (Figure 5a). To reach a more reliable conclusion, we analyzed the expression of mRNA of paired tumor and adjacent nontumor tissues from PC patients (GSE28735, GEPIA and BBCancer databases). Cp expression was significantly enhanced in PC tissues compared to that in adjacent nontumor tissues (Figure 5b). Moreover, because GnT III is responsible for the biosynthesis of bisecting GlcNAc [29], we further analyzed the expression of GnT III in the above three databases. The results indicated that GnT III was also upregulated in PC tissues compared to adjacent nontumor tissues (Figure 5c).

## 4. Discussion

PC is a highly malignant solid tumor and usually demonstrates aggressive metastasis [30]. Its main metastasis process contains detaching from primary cell, invading to tumor stroma, perfusing to blood vessels and extravasating from the organs [31]. Most of the patients with PC are already in the middle or advanced stages at the time of diagnosis, and they have missed the key time point of treatment due to the rapid metastasis of tumor [2]. Early and correct diagnosis of PC is the most urgent issue, even with the aid of modern detection technologies. Many PC-associated serological markers, such as CA19-9, CEA, and fucosylated haptoglobin have been proposed; however, none of them have satisfactory sensitivity and specificity [5,32]. Meanwhile, because AP is reported to be a risk of PC and may constitute to an early symptom of PC [33,34,35,36], some acute-phase proteins that were upregulated in AP [37] were also tested, and the results revealed that some of them increased in PC patients’ sera [38]. Moreover, *N*-glycosylation of serum proteins were dramatically changed in both PC and AP patients [39]. Therefore, in this study, we used AP patients and NC individuals as controls to find a novel and satisfactory molecular biomarker with special focus on cancer-associated glycoalteration involved in PC patients’ sera. Usually, serum is the preferred specimen for the early diagnosis of malignant tumors because of its easy accessibility by less-invasive methods.

In this study, the alteration of glycans in the sera of NC individuals and AP and PC patients were investigated by MS and Lectin blotting. The results of alterations of fucosylation, sialylation and high mannose were consistent with the previous studies on PC cell lines [40] and the serum glycoproteins of PC [41,42]. In terms of glycan structures, Nan B.C. et al. found that the specific activities of GnT III increased in PC as compared with normal pancreatic tissues, and the bisecting GlcNAc structures in *N*-glycans of pancreatic ribonuclease also increased in PC tissues [43]. GnT-III is an important glycosyltransferase that catalyzes the transfer of GlcNAc to the core β-mannose residue of *N*-glycans via the β1,4-linkage to make a bisecting GlcNAc structure in *N*-glycan biosynthetic pathways. The addition of a bisecting GlcNAc by GnT-III to target proteins typically inhibits the action of other glycosyltransferases such as GnT-IV and GnT-V. Therefore, GnT-III can act as a regulator of other glycosyltransferases [44]. Aberrant expression of GnT-III and bisecting GlcNAc has also been observed in various cancer types [15,17,45]. Here, it is the first time that bisecting GlcNAc glycans were reported to be increased in the sera of PC patients. Moreover, PHA-E-positive glycans were chosen as the first probe in biomarker exploration due to its differential expression among PC, AP, and NC groups. Thus, PHA-E was used as a probe to capture the differential proteins, and they were then identified by LC-MS/MS. However, it is noted that the PHA-E recognition is not specific and cannot tell the relative amount of the bisecting GlcNAc structure [46]. Therefore, in this study, we have tried to prove the presence of a bisecting GlcNAc structure in peak 6, 11, 22, 26, and 27 (Figure 2a) by MS/MS, but failed finally due to the low signal intensities of these glycans.

In Figure 2a, sialylated and α1-6 core fucosylated tetra-antennary *N*-glycan was only detected in PC patients’ sera (peak 40), which indicated that this glycan may become a potential biomarker for the diagnosis of PC. In fact, Nouso et al. detected the expression of glycans in the sera of 92 PC patients and 243 NC through glycomics analysis and found that 15 glycans including this glycan were increased in PC, and 10 glycans were significantly upregulated in distant metastasis cases. In the diagnosis respect of pancreatic cancer, this glycan showed the AUC = 0.723 [47].

Besides bisecting GlcNAc glycans, proteins were also selected as biomarker candidates in this study based upon the following characteristics: (1) Protein functions were involved in cancer progression; (2) the MWs of protein candidates are similar to the sizes of the corresponding bands that we cut and analyzed; (3) the glycan structures of the proteins were not previously reported in PC; (4) commercially available antibodies exist for further immunoassays. Among all the PHA-E pulled-down proteins, Cp and Apo-E were chosen for the further analysis of their expression levels in PC, AP, and NC groups. The results indicated that the protein levels of Cp and Apo-E were increased in both AP and PC patients, but there was no obvious difference between PC and AP patients. However, statistical significance was observed in the Cp expression between the AP and PC samples after the proteins were pulled down by PHA-E lectin. Thus, the measurements of PHA-E-positive Cp by Western blotting and lectin blotting would be a more favorable indicator for PC in terms of glycoalteration. In the case of Apo-E, no significant difference was found between the AP and PC samples either in serum or PHA-E captured serum. A previous report showed that the elevation of Apo-E in pancreatic cancer is likely related with the obstructive jaundice rather than Apo-E acting as a direct marker of pancreatic malignancy [48]. In addition to Cp and Apo-E, it was noted that two enhanced bands (at about 25 and 50 kD) were apparently enhanced in the PC group in PHA-E staining (Figure 1b), indicating that PC patients’ sera contained more proteins that bound to PHA-E. Searching glycoproteins with a MW of about 50 kD in Table 2 found that alpha-1-B-glycoprotein (51.9 kD) and alpha1 antitrypsin (45.7 kD) were also related to carcinoma diagnosis [49,50,51,52], which could be further studied for PC diagnosis. Furthermore, since MS-based investigations of glycopeptide was also an ideal tool for biomarker exploration [53], the glycopeptides specific for 3–5 upregulated bisecting modified glycoproteins in PC group identified in this study would also be potential biomarkers for PC diagnosis.

It is well known that most candidate biomarkers for cancer are associated with cellular processes of tumor development and malignant progression. Cp is a member of the multicopper oxidase family of enzymes. Previous reports showed that Cp is not only synthesized in the liver, but also produced by cancer cells [54]. Tumor cells can capture non-Cp copper from plasma to form pathological angiogenesis, which need a relatively large amount of copper; therefore, the expression level of Cp might represent a supportive biomarker for bile duct cancer and breast cancer [54,55]. In our study, we evaluated the diagnostic value of PHA-E-positive Cp for PC by drawing ROC curves. The results showed that PHA-E-positive Cp can predict PC from healthy individuals well with an AUC of 0.972, which was higher than that of CA19-9 (0.86–0.87) [56,57]. It also displayed a relatively satisfactory ability in distinguishing cancer from acute inflammation with an AUC of 0.757, which was slightly higher than CA19-9 in differentiating PC from chronic pancreatitis (0.71) [56]. Moreover, we detected the expression of Cp in PC tissues by IHC and analyzed the existing data in databases and found that Cp was upregulated in PC tissues. All these results indicated a close relationship between Cp generation and PC development and implied a potential value of PHA-E-positive Cp in the diagnosis of PC. However, a quantitative and high-throughput ELISA-format system is more preferred for the detection of PHA-E-positive Cp clinically. In this ELISA system, PHA-E can be immobilized in 96-well plates to capture PC-specific glycan moiety, and Cp antibody can be used as the detection probe.

In summary, our findings showed that PHA-E-positive Cp was overexpressed in PC patients’ sera and tumor tissues. Using PHA-E and anti-Cp antibody as probes, PC could be distinguished from NC and AP patients, especially from NC individuals. Therefore, PHA-E-positive Cp is a potential biomarker for the diagnosis of PC, which needs to be validated and researched further for clinical use.

## Figures and Tables

**Figure 1 cells-11-02453-f001:**
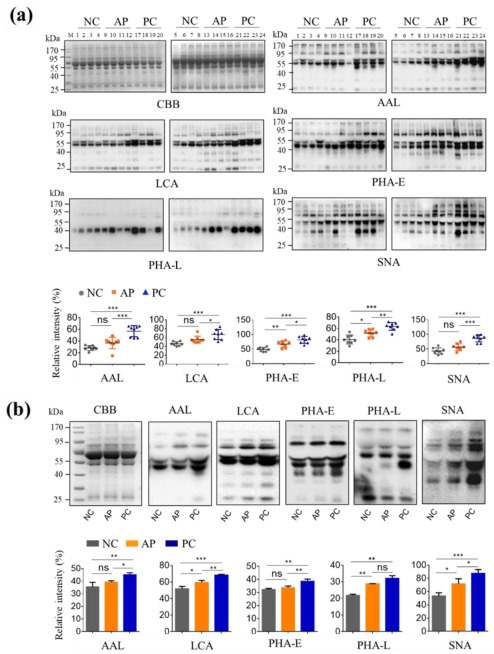
Lectin blotting of PC patients’ serum glycoproteins. (**a**) A total of 5 μg of serum proteins from NC individuals (1–8), AP (9–16), and PC (17–24) patients were subjected to 10% SDS-PAGE, stained with Coomassie brilliant blue (CBB), or electroblotted membranes to stain with AAL, LCA, PHA-E, PHA-L, and SNA lectins. (**b**) A total of 5 μg of proteins of pooled sera from NC (*n* = 8), AP (*n* = 8), and PC (*n* = 8) groups were subjected to 10% SDS-PAGE and stained with CBB, and other electroblotted membranes were stained with above lectins. NC, normal control; AP, acute pancreatitis; PC, pancreatic cancer. AAL binds to α1-2/3/4/6-linked fucose residues. LCA binds to α1-6-linked fucose residue. PHA-E binds to bisecting GlcNAc *N*-glycans. PHA-L binds to β1,6 GlcNAc-branched *N*-glycans. SNA binds to α2-6-linked sialic acid residues. The relative intensities of the lectin staining were analyzed using ImageJ, normalized with total protein signal in CBB staining, and finally statistically analyzed by Graphpad Prism 6.0. All values are means ± SD (error bars) from 8 individuals of one group (**a**), or from three independent experiments (**b**) *, significantly different *p* < 0.05; **, *p* < 0.01; ***, *p* < 0.001; ns, no significant difference.

**Figure 2 cells-11-02453-f002:**
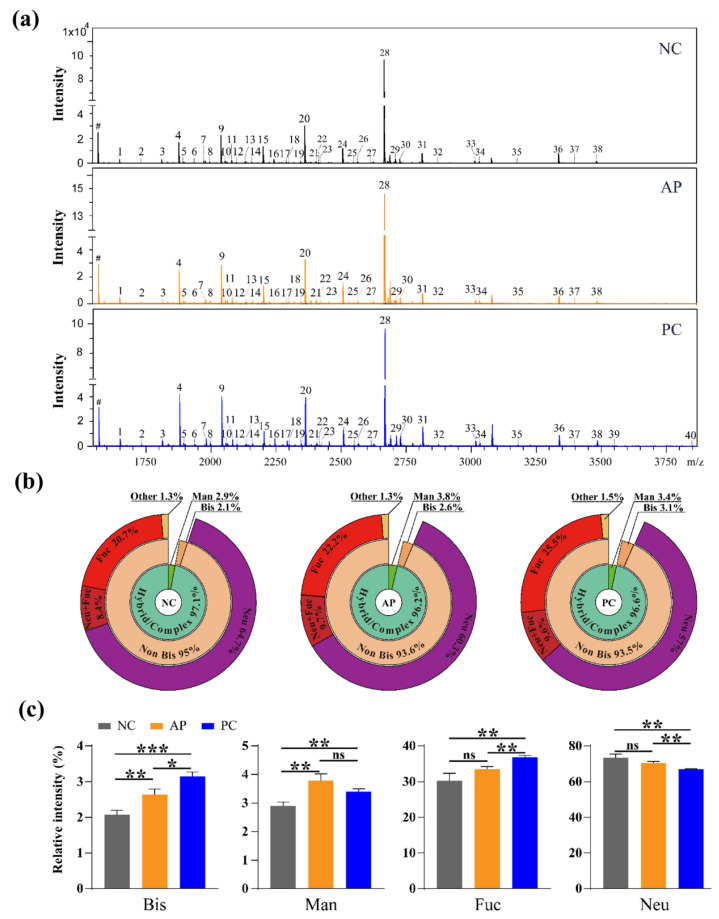
*N*-Glycans analysis of PC patients’ serum glycoproteins by MS. (**a**) MALDI-TOF MS spectra of *N*-glycans of pooled human sera derived from NC (top, *n* = 8), AP (middle, *n* = 8), and PC (bottom, *n* = 8). (**b**) Sunburst charts showing the relative abundances of all the *N*-glycan types in different groups. (**c**) Relative intensities of bisected, mannosylated, fucosylated, and sialylated *N*-glycans were statistically analyzed. The relative intensity of each glycan was calculated by normalizing the intensity of an individual glycan with the intensity of internal standard glycan. # indicates the internal standard glycan. Glycan peaks are numbered and summarized in Table 1. All values are means ± SD (error bars). NC, normal control; AP, acute pancreatitis; PC, pancreatic cancer; *, significantly different *p* < 0.05; **, *p* < 0.01; ***, *p* < 0.001; ns, no significant difference.

**Figure 3 cells-11-02453-f003:**
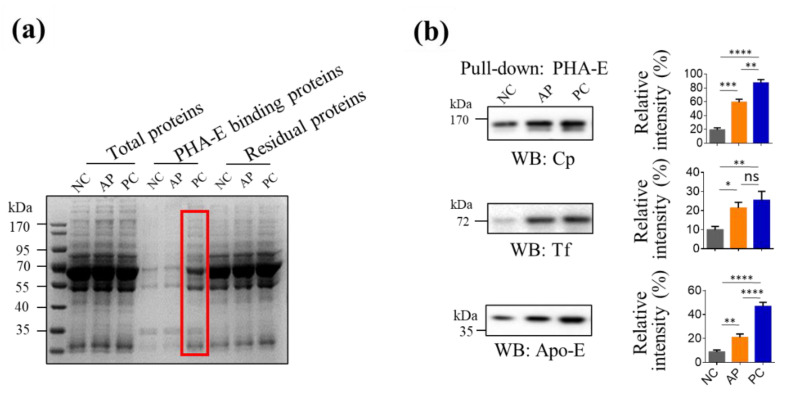
Detection of bisecting glycoproteins in PC patients’ sera. (**a**) PHA-E lectin pull-down and SDS-PAGE analysis. Serum proteins from NC, AP, and PC groups were treated with TCA to remove albumins and then 350 μg proteins were employed to perform pull-down experiment with PHA-E overnight at 4 °C. Then, the mixture was incubated with streptavidin-agarose and the exacted proteins and residual proteins were analyzed by SDS-PAGE. An amount of 6 μg of total proteins, 20 μL of PHA-E binding proteins, and 6 μg of residual proteins for each sample was loaded onto the gel. Protein bands differently expressed in PC group as compared to NC and AP groups (indicated by red frame) were cut and extracted for LC-MS analysis. (**b**) The expression levels of Cp, Tf, and Apo-E in PHA-E-precipitated proteins among pooled sera of NC, AP, and PC were analyzed by Western blotting (left). Twenty milliliters of PHA-E-precipitated proteins of each group was loaded onto the gel. The relative intensities of bands were analyzed by Image J software, normalized with total protein signal in CBB staining of the first three lines in (a), and finally statistically analyzed by Graphpad Prism 6.0 (right). The values are means ± SD (error bars) from three independent experiments. NC, normal control; AP, acute pancreatitis; PC, pancreatic cancer; Cp, ceruloplasmin; Tf, transferrin; Apo-E, apolipoprotein E; WB, Western blotting; *, *p* < 0.05; **, *p* < 0.01; ***, *p* < 0.001; ****, *p* < 0.0001; ns, no significant difference.

**Figure 4 cells-11-02453-f004:**
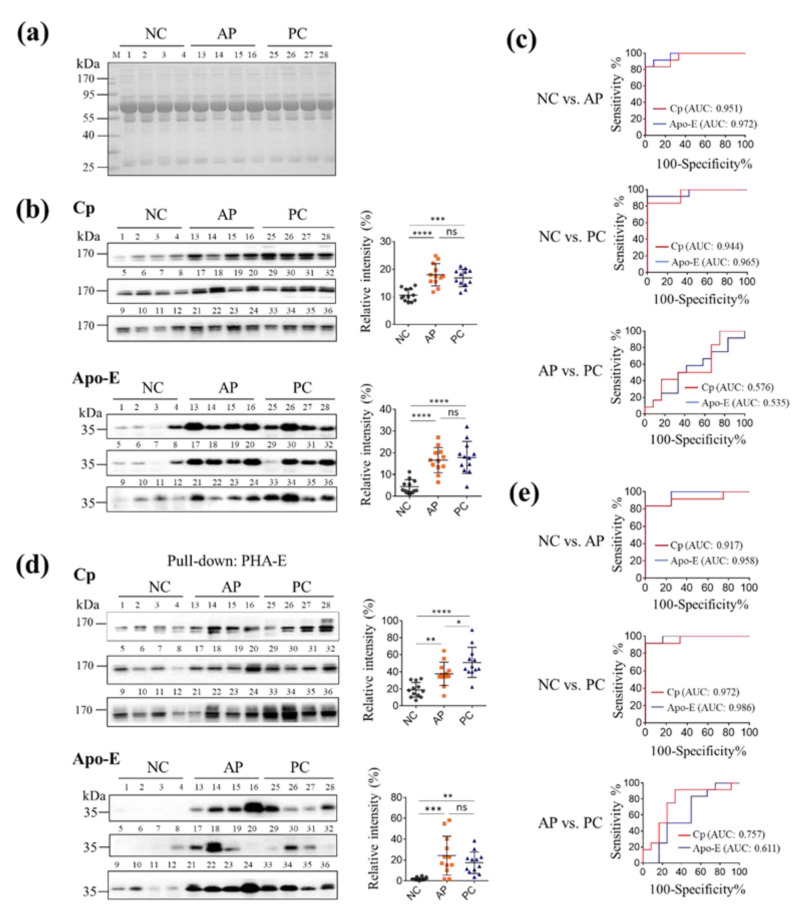
The expression of Cp and Apo-E and the expression of PHA-E-positive Cp and Apo-E. The protein concentrations of sera from 12 NC individuals (1–12), 12 AP (13–24), and 12 PC (25–36) patients were analyzed by CBB staining. The results of one gel were shown in (**a**), and another two gels can be seen in Appendix A. (**b**) Detection of the expression of Cp and Apo-E in the sera by Western blotting. An amount of 6 μg of serum proteins was applied for each sample. (**c**) Evaluation of serum Cp and Apo-E in diagnosis of AP and PC by drawing ROC curves. (**d**) Detection of the level of Cp and Apo-E in PHA-E pulled down sera by Western blotting. A total of 350 μg serum proteins for each sample was used in PHA-E pull-down experiment and 20 μL pulled-down serum proteins was loaded onto the gel. (**e**) Evaluation of Cp and Apo-E in PHA-E pulled-down sera in diagnosis of AP and PC by drawing ROC curves. The relative intensities of bands (**b**,**d**) were analyzed using Image J software, normalized with total protein signal in CBB staining (a and Appendix A), and finally statistically analyzed by Graphpad Prism 6.0. All values are means ± SD from 12 individuals of one group. NC, normal control; AP, acute pancreatitis; PC, pancreatic cancer; Cp, ceruloplasmin; Apo-E, apolipoprotein E; *, *p* < 0.05; **, *p* < 0.01; ***, *p* < 0.001; ****, *p* < 0.0001; ns, no significant difference.

**Figure 5 cells-11-02453-f005:**
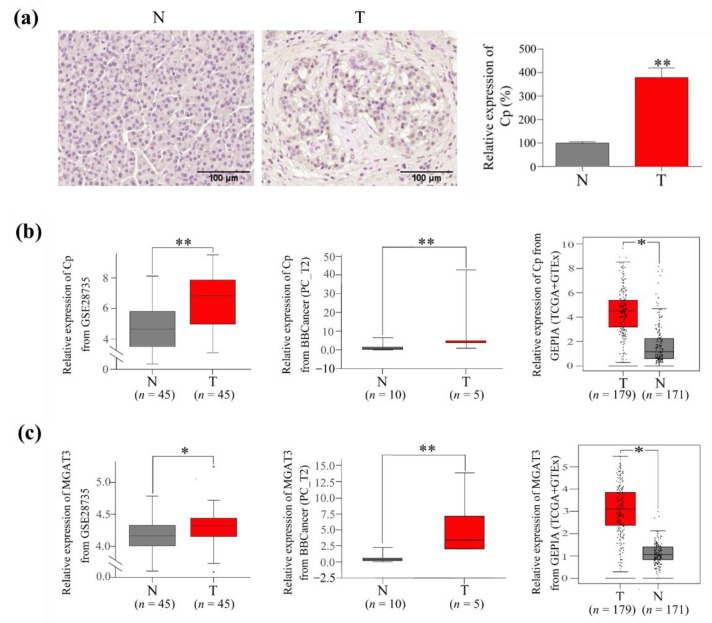
The expression of Cp and GnT III in PC tissues. (**a**) The expression of Cp in paired PC tissue and adjacent normal pancreatic tissue were analyzed by immunohistochemistry (IHC). Results represent the means ± SD (*n* = 3). (**b**) The expression of Cp in PC tissue and adjacent nontumor tissue. (**c**) The expression of GnT III (MGAT3) in PC tissue and adjacent nontumor tissue. The raw data were downloaded from GSE28735, GEPIA and BBCancer databases. Cp, ceruloplasmin; N, adjacent normal pancreatic tissue; T, PC tissue; *, *p* < 0.05; **, *p* < 0.01.

**Table 1 cells-11-02453-t001:** Summary of *N*-linked glycans released from the sera of NC individuals and AP and PC patients and identified by MALDI-TOF MS.

Peak	Calculated *m/z* ^(a)^	Chemical Composition ^(b)^	Relative Intensity (%)	Structure
No.	NC	AP	PC
1	1649.67	(Hex)_2_ + (Man)_3_(GlcNAc)_2_	1.18	1.62	1.44	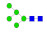
2	1731.72	(HexNAc)_2_ + (Man)_3_(GlcNAc)_2_	0.13	0.21	0.28	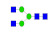
3	1811.72	(Hex)_3_ + (Man)_3_(GlcNAc)_2_	1.08	1.31	1.17	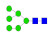
4	1877.78	(HexNAc)_2_(Fuc)_1_ + (Man)_3_(GlcNAc)_2_	6.31	8.05	10.97	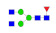
5	1893.77	(Hex)_1_(HexNAc)_2_ + (Man)_3_(GlcNAc)_2_	0.37	0.43	0.49	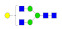
6	1934.80	(HexNAc)_3_ + (Man)_3_(GlcNAc)_2_	0.13	0.17	0.38	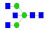
7	1973.77	(Hex)_4_ + (Man)_3_(GlcNAc)_2_	0.18	0.26	0.25	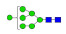
8	1995.78	(Hex)_1_ (HexNAc)_1_ (NeuAc)_1_ + (Man)_3_(GlcNAc)_2_	0.41	0.47	0.53	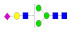
9	2039.83	(Hex)_1_(HexNAc)_2_(Fuc)_1_ + (Man)_3_(GlcNAc)_2_	8.45	9.19	10.15	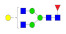
10	2055.83	(Hex)_2_(HexNAc)_2_ + (Man)_3_(GlcNAc)_2_	0.69	0.57	0.51	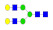
11	2080.86	(HexNAc)_3_(Fuc)_1_ + (Man)_3_(GlcNAc)_2_	0.74	1.07	1.21	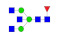
12	2096.85	(Hex)_1_(HexNAc)_3_ + (Man)_3_(GlcNAc)_2_	0.14	0.19	0.26	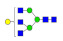
13	2135.82	(Hex)_5_ + (Man)_3_(GlcNAc)_2_	0.23	0.34	0.27	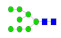
14	2157.83	(Hex)_2_(HexNAc)_1_(NeuAc)_1_ + (Man)_3_(GlcNAc)_2_	0.25	0.42	0.35	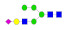
15	2201.88	(Hex)_2_(HexNAc)_2_(Fuc)_1_ + (Man)_3_(GlcNAc)_2_	5.00	4.00	3.01	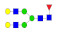
16	2242.91	(Hex)_1_(HexNAc)_3_(Fuc)_1_ + (Man)_3_(GlcNAc)_2_	0.90	1.01	1.32	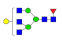
17	2297.88	(Hex)_6_ + (Man)_3_(GlcNAc)_2_	0.21	0.26	0.27	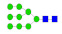
18	2319.89	(Hex)_3_(HexNAc)_1_(NeuAc)_1_ + (Man)_3_(GlcNAc)_2_	0.16	0.31	0.26	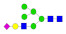
19	2344.92	(Hex)_1_(HexNAc)_2_(Fuc)_1_(NeuAc)_1_ + (Man)_3_(GlcNAc)_2_	0.29	0.3	0.38	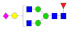
20	2360.91	(Hex)_2_(HexNAc)_2_(NeuAc)_1_ + (Man)_3_(GlcNAc)_2_	10.93	10.50	8.60	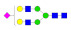
21	2401.94	(Hex)_1_(HexNAc)_3_(NeuAc)_1_ + (Man)_3_(GlcNAc)_2_	0.14	0.20	0.25	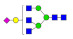
22	2404.96	(Hex)_2_(HexNAc)_3_(Fuc)_1_ + (Man)3(GlcNAc)2	0.43	0.50	0.52	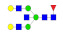
23	2420.96	(Hex)_3_(HexNAc)_3_ + (Man)_3_(GlcNAc)_2_	0.12	0.13	0.18	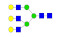
24	2506.97	(Hex)_2_(HexNAc)_2_(Fuc)_1_(NeuAc)_1_ + (Man)_3_(GlcNAc)_2_	3.90	4.03	3.10	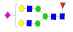
25	2548.00	(Hex)_1_(HexNAc)_3_(Fuc)_1_(NeuAc)_1_ + (Man)_3_(GlcNAc)_2_	0.08	0.14	0.20	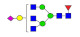
26	2563.99	(Hex)_2_(HexNAc)_3_(NeuAc)_1_ + (Man)_3_(GlcNAc)_2_	0.31	0.38	0.38	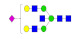
27	2624.04	(Hex)_3_(HexNAc)_4_ + (Man)_3_(GlcNAc)_2_	0.32	0.33	0.39	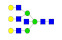
28	2666.01	(Hex)_2_(HexNAc)_2_(NeuAc)_2_ + (Man)_3_(GlcNAc)_2_	49.77	45.42	44.14	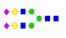
29	2710.05	(Hex)_2_(HexNAc)_3_(Fuc)_1_(NeuAc)_1_ + (Man)_3_(GlcNAc)_2_	1.36	1.73	1.60	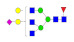
30	2726.05	(Hex)_3_(HexNAc)_3_(NeuAc)_1_ + (Man)_3_(GlcNAc)_2_	1.02	0.96	1.34	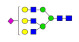
31	2812.07	(Hex)_2_(HexNAc)_2_(Fuc)_1_(NeuAc)_2_ + (Man)_3_(GlcNAc)_2_	2.04	2.55	2.67	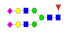
32	2872.10	(Hex)_3_(HexNAc)_3_(Fuc)_1_(NeuAc)_1_ + (Man)_3_(GlcNAc)_2_	0.11	0.13	0.19	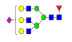
33	3015.15	(Hex)_2_(HexNAc)_3_(Fuc)_1_(NeuAc)_2_ + (Man)_3_(GlcNAc)_2_	0.32	0.48	0.65	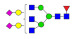
34	3031.14	(Hex)_3_(HexNAc)_3_(NeuAc)_2_ + (Man)_3_(GlcNAc)_2_	0.37	0.32	0.36	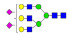
35	3177.20	(Hex)_3_(HexNAc)_3_(Fuc)_1_(NeuAc)_2_ + (Man)_3_(GlcNAc)_2_	0.06	0.06	0.13	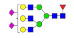
36	3336.24	(Hex)_3_(HexNAc)_3_(NeuAc)_3_ + (Man)_3_(GlcNAc)_2_	1.55	1.65	1.06	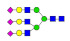
37	3396.27	(Hex)_4_(HexNAc)_4_ (NeuAc)_2_ + (Man)_3_(GlcNAc)_2_	0.05	0.05	0.07	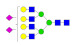
38	3482.29	(Hex)_3_(HexNAc)_3_(Fuc)_1_(NeuAc)_3_ + (Man)_3_(GlcNAc)_2_	0.23	0.26	0.58	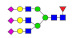
39	3542.33	(Hex)_4_(HexNAc)_4_ (Fuc)_1_ (NeuAc)_2_ + (Man)_3_(GlcNAc)_2_	—	0.00	0.05	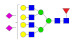
40	3847.43	(Hex)_4_(HexNAc)_4_ (Fuc)_1_ (NeuAc)_3_ + (Man)_3_(GlcNAc)_2_	—	0.00	0.04	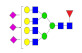

*m*/*z* values are monoisotopic; “—” indicates “not detected”; NC, normal control; AP, acute pancreatitis; PC, pancreatic cancer. ^(a)^ The glycans were calculated as aminooxy tryptophanylarginine-labeled derivatives in which neuraminic acids were methylesterified, and [M+H]^+^. ^(b)^ Monosaccharide compositions were determined by database searching using GlycoMod (http://www.expasy.ch/ tools/glycomod/, accessed on 20 December 2021). Monosaccharides are indicated as follows: Hex, hexose; HexNAc, *N*-acetylhexosamine; Fuc, fucose ▼; GlcNAc, *N*-acetylglucosamine ■; NeuAc, *N*-acetylneuraminic acid ♦; Man, mannose ●; Gal, Galactose ●.

**Table 2 cells-11-02453-t002:** Identification of glycoproteins with bisecting GlcNAc in PC patients’ sera.

Protein Name	Accession	Unique Peptides	Score	Coverage	mol wt (kDa)	PI
chain B, Human Complement Component C3	gi|78101268	30	1131.09	31.5%	112.9	5.46
keratin 1	gi|11935049	11	1057.63	29.7%	66	8.83
keratin 10	gi|21961605	9	1034.71	38.5%	58.8	4.95
complement C3 precursor	gi|115298678	27	1003.96	18.2%	187	5.98
keratin, type II cytoskeletal 6A	gi|5031839	21	885.6	34.9%	60	8.89
type I keratin 16	gi|1195531	17	875.29	36.4%	51.2	4.84
keratin type II	gi|908790	3	847.97	34.0%	60	8.89
keratin 1	gi|7331218	11	666.54	20.3%	66	8.83
immunoglobulin light chain	gi|149673887	7	650.28	63.3%	23.4	7.62
chain A, Alpha1-Antitrypsin	gi|157831596	13	642.67	43.9%	44.2	5.27
human apolipoprotein A-I	gi|90108664	12	615.34	51.9%	28.1	5.15
alpha-1-antichymotrypsin precursor, partial	gi|177933	14	598.47	36.1%	45.5	5.21
immunoglobulin kappa light chain VLJ region	gi|21669395	9	582.35	41.5%	30.5	9.34
keratin K5	gi|18999435	13	539.06	21.7%	62.3	5.5
cytokeratin 9	gi|435476	11	490.09	23.0%	62.1	5.06
serum transferrin	gi|110590597	12	459.46	18.2%	74.7	6.61
unnamed protein product	gi|34527290	8	450.68	23.2%	53.2	5.46
plasma protease (C1) inhibitor precursor	gi|179619	5	399.48	19.6%	55.1	5.09
complement factor B	gi|291922	10	375.61	14.7%	85.6	6.59
unnamed protein product	gi|28317	8	371.36	15.7%	59.5	5.04
ALB	gi|37222202	8	367.55	57.5%	19.3	5.14
haptoglobin precursor	gi|306882	7	362.64	15.8%	45.8	6.26
immunoglobulin alpha-2 heavy chain	gi|184761	6	342.37	19.2%	36.4	5.68
epidermal cytokeratin 2	gi|181402	8	341.02	14.0%	65.8	8.85
ceruloplasmin	gi|1620909	8	339.04	9.6%	115.4	5.35
immunoglobulin heavy chain constant region	gi|10799664	8	312.54	19.6%	35.9	8.8
Ig A1 Bur	gi|229585	5	310.32	7.4%	73.3	10.09
Ig A L	gi|229536	5	307.89	24.5%	22.8	9.79
haptoglobin Hp2	gi|223976	5	284.97	10.1%	41.7	6.25
immunoglobulin heavy chain	gi|10334587	6	277.133	15.6%	41.3	9.5
hypothetical protein	gi|51476390	6	271.58	10.2%	69.4	5.84
alpha-2-macroglobulin precursor	gi|177870	6	254.79	5.2%	163.2	5.98
Ig G1 H Nie	gi|229601	4	240.54	25.4%	49.2	9.59
serum albumin	gi|28592	32	239.86	10.5%	69.3	6.04
apolipoprotein E	gi|178849	5	235.07	19.6%	36.2	5.53
complement component C5, partial	gi|179692	5	233.34	4.2%	141.7	6.51
apo-B100 precursor	gi|28780	12	222.07	1.7%	514.9	6.68
Ig heavy chain V-III region (ART)	gi|106482	4	208.47	24.8%	23.7	5.04
alpha1 antitrypsin	gi|225768	5	182.04	14.9%	45.7	6.1
immunoglobulin heavy chain constant region	gi|10799664	5	169.05	16.0%	35.9	8.8
alpha-2-macroglobulin precursor	gi|177870	3	128.07	2.7%	163.2	5.98
cytokeratin 9	gi|435476	2	127.28	4.7%	62.1	5.06
hemoglobin beta	gi|229149	2	70.83	18.5%	15.9	5.14
immunoglobulin heavy chain variable region	gi|114147353	2	66.97	16.1%	12.8	10.17
complement component C3	gi|179665	5	60.7	2.5%	187	5.46
alpha-1-B-glycoprotein	gi|69990	2	50.72	6.3%	51.9	5.6
alpha-1 antitrypsin	gi|28637	2	41.58	8.6%	22.8	6.1

## Data Availability

Not applicable.

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
