# Peer review of "Phaseolus vulgaris Erythroagglutinin (PHA-E)-Positive Ceruloplasmin Acts as a Potential Biomarker in Pancreatic Cancer Diagnosis"

_cells, 2022, doi:10.3390/cells11152453_

Round 1

Reviewer 1 Report

The manuscript Bisecting GlcNAc Modified Ceruloplasmin Acts as a Potential Biomarker in Pancreatic Cancer Diagnosisaddresses an important health challenge in pancreatic cancer that is to find biomarkers for early diagnosis.

The authors claim that ceruloplasmin (Cp) carrying bisecting GlcNAc could be a potential biomarker for pancreatic cancer diagnosis.

The analyses of specific glycoforms of serum proteins as a strategy to find new biomarkers is an interesting approach, but in this study I have some concerns regarding the robustness of the findings described:

1. The authors use three groups of study: normal healthy individuals (NC), acute pancreatitis (AP) and pancreatic cancer (PC). Why do they use the acute pancreatitis group? Usually as a benign pancreatic disease, chronic pancreatitis is included as a control group instead of AP, since there is not a clinical problem to distinguish AP from PC.

2. They analyze by lectin blotting the potential of several lectins to differentiate the serum protein pattern of the three groups of study (normal controls (NC), acute pancreatitis (AP) and pancreatic cancer (PC)). The authors should consider that Western Blotting is not a quantitative method and it may show trends in staining. In this case an increasing PHA-E staining in the PC group.

3. When they try to determine the proteins that are stained with PHA-E and therefore that carry bisecting GlcNAc, they pulled down these proteins with PHA E lectin (Figure 3a) and then identified them by MS (Table 2). Many proteins are identified, including ceruloplasmin (Cp). Then they performed WB against Cp on the PHA-E pulled down proteins and detected Cp at a band of 170 kDa (Figure 3b). Then, they quantify the Cp bands of the WB of the PHA-E pulled down proteins (Figure 4d). This technique is semi-quantitative, and to my opinion a quantitative method to quantify Cp carrying bisecting GlcNAc (for instance in an Elisa-format) would be more sound to claim that bisecting GlcNAc is a potential biomarker for PC.

5. Cp is an acute phase proteins and therefore is increased in the AP group.

6. How is the Cp staining in the IHQ quantified?

7. The working concentration of lectins is expressed at 1/15000 or 1/2000 or 1/1000 dilution. Some of these dilution factors appear to be very high. Could they specify the working dilution of the lectins at μg/mL?

Reviewer 2 Report

Shanshan Sha and coworkers have found via lectin-enrichment that ceruloplasmin with bisected GlcNAc could be a diagnostic marker for pancreatic cancer (PC).
Now, PC is indeed a particularly nasty type of cancer with a very poor prognosis once diagnosed. Reliable markers for early diagnosis hence would be highly welcome. Does the current study help to find such a marker?

Serum proteins were analyzed by lectin blotting and PHA-E was selected for further pull-down experiments as it was the most discriminating lectin (even though Fig. 1 does not give a clear preference for this decision.
Fig. 1 indicates that PC sera contain more protein that bound to PHA-E. Among the several bands, three proteins were considered in detail. Of these, the authors finally favored ceruloplasmin. 2 other proteins seem to be enriched as well (ca. 50 and 20 kDa) but are not mentioned.

(Btw. Table 1 – in the absence of any quantitative information  - is a perfectly pointless listing of serum glycans. The relevant data is presented in Fig. 2 anyway)

Finally (almost) the levels of Cp and Apo-E were directly measured by Western blotting involving 12 sera of each group. Differences were found between NC (control) and “inflamed” tissue (AP and PC), whereby the clearest results were obtained after PHA-E fractionation. So it seems that both the expression levels AND the glycosylation in general are changing in PC. This would imply that several proteins are – in a somewhat unusual manner – expressed in PC cells AND these cells equip their products with a clearly different glyco-pattern.
This is an interesting finding and in my eyes – as the case is of high importance – the other proteins blinking up in Fig. 1 (PHA-E pull down lance PC) should receive a similar attention as Cp in Fig. 5.
I suggest that the authors rethink their results under this double-reason light.

What I painfully miss is an experiment that  - for me – would be the ideal combination of all these findings: can glycopeptides specific for the 3-5 upregulated bisecto-proteins be found in LC-MS ? This could be a more practicable way of diagnosis than lectin fractionation plus Western blotting for >1 protein. Details:

The text could well take some language editing:
Example: l 14: … satisfied reliability  

Give a source for the PHA-E specifity

Reviewer 3 Report

The article focuses on the differential glycosylation of glycoproteins in the health pancreatic tissues with respect to pancreatitis and cancer, mainly studied by lectin reactivity. The issue is interesting but the data presentation is not convincing and requires more rigorous procedures.

Lectin blots are presented as percent of relative intensity, but it is not clear what means. It is necessary to state the value corresponding to 100. The amounts of protein loaded is of course identical between samples, but an internal standard for blotting normalization is still necessary. The authors should include the visualization of a reference standard and perform calculations accordingly. Moreover and more important, the apparent reactivity of single cases of the three groups is wide, as usual between human samples, but this is not represented in the standard deviations. The meaning of pooling samples is not clear and should be limited to the MS experiments or carefully explained. To establish the statistical significance the authors should consider the means for the 8 individual cases of each group, calculate the related standard deviations, and then they can define whether the differences between groups are statistically significant. The statistics of the technical replicates is not relevant in this case. Moreover, signals from several sample appear oversaturating in a way that impairs correct calculations and should be repeated after proper dilutions.

Minor

CA19.9 should be written homogeneously thoughout the text

Reviewer 4 Report

The manuscript of Shanshan Sha et al. entitled “Bisecting GlcNAc Modified Ceruloplasmin Acts as a Potential Biomarker in Pancreatic Cancer Diagnosis presents a very valuable study to improve the diagnosis of pancreatic cancer. The authors use state of the art methodology to study N-glycan compositions and expression levels of a wide range of serum proteins in cohorts of patients with pancreatic cancer and reference cohorts. Two potential marker proteins were found to be significantly increased in patients with pancreatic cancer compared to healthy individuals: ceruloplasmin (Cp) and Apo-E. Both have significant amounts of bisecting GlcNAcs structures and can be enriched with the lectin PHA-E. Immunohistochemistry showed that Cp is also significantly higher expressed in tumor tissue of PC patients than in adjacent normal pancreatic tissue.

In my view the most important finding is that increased levels of bisecting GlcNAcs were found in sera of pancreatic cancer-patients, both analyzed as bulk N-glycans and also on the individual serum proteins Cp and Apo-E. Very interesting are the statistical data of cohorts of PC patients, healthy individuals and patients with acute pancreatitis. The PC patients show significantly higher expression of the proteins Cp and Apo-E. However, this is also the case for patients with acute pancreatitis (AP). The data for PC patients and AP cannot be distinguished based on these expression levels. Only after a pull-down with PHA-E the observed intensities for PC and AP cohorts show a significant difference for the expression of the Cp protein (but not for Apo-E). This is quite an important finding.

1. The graphs of the pull-down glycoproteins with PHA-E lectin are very convincing, especially for the proteins Cp and Apo-E. The pancreatic cancer (PC) patients show significantly stronger intensities than healthy persons and those with acute pancreatitis. Patients with acute pancreatitis show especially on Cp significantly larger intensities than healthy patients.

For the protein Tf the data in Fig. 3 b is less convincing. Although the pancreatic cancer-patients show more intensity than patients with acute pancreatitis in the graph, a visual inspection of the Western Blot seems to me that the band of the AP patients is stronger.

2. The significant difference between the observed intensities of Cp for the PC and AP cohorts (after a pull-down with PHA-E) is quite an important finding. However, the individual data distributions of AP and PC overlap and although there is a significant difference, the area under the ROC curve (AUC) is with 0.757 quite promising but not excellent. An AUC or 0.757 means 75.7% correct prediction and 24.3% incorrect prediction.

With these values in mind, I think the conclusion (in line 424) that “PC could be well distinguished from NC and AP patients” is a bit overstated.

I fully agree that PC patients can be well distinguished from NC, but the distinction between PC and AP patients is not so clear.

Nevertheless, the clear distinction between PC and NC is a very valuable finding, even if patients with acute pancreatitis are not so clearly distinguishable from PC. Maybe, even a straightforward diagnostic distinction between PC+AP from healthy individuals is useful, I guess there are also other ways to diagnose acute pancreatitis and to distinguish this condition from PC.

3. The authors should explain a bit better the connection between GnT III and their finding. They mention GnT III, GnT IV and GnT V (together) in the context of the study by Nan et al. 1998. Only the N-acetylglucosaminyltransferases GnT III is responsible for generating bisecting GlcNAc structures, which is the focus of the current paper. This should be better explained.

4. Some abbreviations are a bit difficult to remember and their origin is not so clear, e.g. NC standing for “normal healthy individuals”. I wonder what the C is standing for. Maybe cohort?

Another example is aoWR, which should stand for aminooxy-functionalized peptide reagent. What does the W stand for?

Some abbreviations are not explained: TBST, TCA, BCA

As also mentioned in the introduction there are already studies/hints in the literature that showed elevated bisecting GlcNAcs in other types of cancer, e.g. in fibronectin from urine of patients with bladder cancer (Guo et al. 2001, J. Cancer Res. Clin. Oncol 127, 512). However, these studies support the significance of bisecting GlcNAcs in cancer, but they did not study it in such detail like the current manuscript and did not provide statistics of cohorts of patients + healthy individuals. Therefore, I believe this manuscript is a very important and valuable contribution!

In some occasions the English should be improved:

line 77: beginning the sentence with “Previous study … revealed …” should be “A previous study …”

line 387: “…and found that both the activities of GnT III, GnT IV and 387 GnT V and the antenna and bisecting GlcNAc structures in N-glycans …”. Here it is not clear what both activities refer to. Do all the enzymes have two activities? There seem to be too many “and”.

line 393: “… proteins, which then identified by 393 LC-MS/MS”. Proteins cannot identify themselves. I guess “they were then identified”

line 397: “… involved in a cancer progression”. The “a” does not fit there.

line 408: "exhibited no differential indication" is not an English expression, I cannot understand the meaning.

line 419: “in clinical database”. In a specific one? Then please name it. Or do you mean “in clinical databases” (plural).

line 114: I guess in “dl-dithiothreitol” the “dl” describes the chirality and then both letters should be written with small caps font.

Reviewer 5 Report

In this manuscript, the authors highlighted ceruloplasmin carrying bisecting GlcNAc N-glycan as a potential glyco-biomarker to discriminate pancreatic ductal adenocarcinoma (PDAC) patients from acute pancreatitis patients and healthy subjects. This study is  interesting and the results are well presented. However there are a number of issues with the manuscript that the author should consider :

- It is unfortunate that this study was performed only on acute pancreatitis patients. Why not have used sera from chronic pancreatitis patients ? Chronic pancreatitis is an important inflammatory pancreatic pathology and a risk factor for PDAC. If possible, sera from these patients should be included in this study ! Otherwise this point should be discussed.

- Pancreatic ductal adenocarcinoma is more accurate than pancreatic cancer which includes different types of cancer of pancreas. The authors must specify in the manuscript otherwise they must give the types of pancreatic cancer studied in the text of manuscript and table S1.

- There is little information about clinical data of patients in table S1 in particular for PDAC patients. More informations are needeed (type of pancreatic cancer : PDAC or other ; disease stage : localized, locally advanced, metastatic ; resected or unresectable tumors ; histological and anatomo-pathological data of tissues ; number of patient correlated with number of sample presented in western-blot ; … ).

- The meaning of « CBB » in figure 1 should be added in the legend of figure.

- There is no information about normalization of proteins used to determine the relative intensities of proteins studied (lectins and specific antibodies) of western blotting data; more technical details are needed (relative expression normalized with total protein signal stained with CBB ?).

- Regarding the figure 1 (and figure S1) the numbering of the samples is confusing; it would be better to number the serum protein samples from NC subjects from 1 to 8, from AP patients from 9 to16, etc… on the 2 western-blots (correlated with the numbering in table S1, see above).

- The figure 2b are too small to be correctly analysed, specifically the percentage and name of each sub-population of N-glycans is illegible. It is necessary to enlarge them to increase their clarities.

- Regarding the figure 3A and B, the amount of proteins loaded on SDS-PAGE should be given in the legend of figure.

- Regarding the results presented in Figure 2a, the authors have observed sialylated and a(1-6) core-fucosylated tetra-antennary N-glycans only in PDAC patient serum (peak n° 40). This result is very interesting but unfortunatly, it was neither discussed nor exploited as a potential biomarker for PDAC diagnosis in this manuscript. This point should be clarified and discussed.

- Regarding the figure 5a: the magnifications seems to be different between tumoral tissue and adjacent normal tissue. This point should be corrected. The scale bars should be given. It would be more interesting to show the Cp expression on a picture showing the border between tumoral area and adjacent normal area of PDAC tissues. How the relative expression of Cp is calculated from the IHC ? This point should be clarified. Regarding the figure 5b, black color of box-plots should be changed, it is not suitable to see the median in the box.

- To consolidate the results of Cp expression obtained from database, the expression of B6GNT3 involved in the formation of bisecting N-glycan should be determined from the 3 GEO database.

- Overall, why the authors did not use ELISA, a more robust technique to quantify Cp expression after PHA-E pull-down rather than. This point should be clarified. If possible, ELISA should be performed to confirm the western-blot data obtained.

- Many typographical errors must be corrected : e.g. : « alteration or alternation ? » ; « CA199 »; « individuials » ; « glycol-alteration » ; …

Round 2

Reviewer 1 Report

The authors have partially addressed the suggestions. They have included the explanations of the WB and IHC quantifications, but some of the main concerns could not be properly addressed, and overall there is still room to improve the quality of the manuscript.

They have justified why they did not include chronic pancreatitis patients, although they expect to do it, which would improve the quality of the manuscript. In addition, they have explained that they tried to quantify the bisecting GlcNAc on ceruloplasmin by a quantitative method such as ELISA, but they could not continue on that and will do it in the future. Overall, this makes that the results presented in this version are preliminary and therefore further validation by a quantitative technique is required.

These aspects would need extra-time to be addressed but they would add value to the manuscript and to the aim of the study that is to describe bisecting GlcNAc modified ceruloplasmin as a potential biomarker in pancreatic cancer diagnosis

Reviewer 3 Report

The authors rebuttals and article modifications are scientifically satisfactory. In the revised text the English style should be improved. Examples:

Equal quality of serum/ quantity?

alteration of PC patients’ serum proteins/ unclear.

punctuation in seveeral parts.

Reviewer 5 Report

None

Author Response

Thank you very much.